# Serum Levels of MMP-8 and MMP-9 as Markers in Chronic Subdural Hematoma

**DOI:** 10.3390/jcm11040902

**Published:** 2022-02-09

**Authors:** Gao-Jian Su, Jie Gao, Chu-Wei Wu, Jun-Feng Zou, Di Zhang, Dong-Liang Zhu, Jun Liu, Jie-Hua Zhang, Xian-Jian Huang

**Affiliations:** Shenzhen Key Laboratory of Neurosurgery, Department of Neurosurgery, the First Affiliated Hospital of Shenzhen University, Shenzhen Second People’s Hospital, Shenzhen 518000, China; 14gjsu1@stu.edu.cn (G.-J.S.); 17zychen9@stu.edu.cn (J.G.); 17shlao@stu.edu.cn (C.-W.W.); 1810249001@email.szu.edu.cn (J.-F.Z.); 2070245015@email.szu.edu.cn (D.Z.); 14dwrui1@stu.edu.cn (D.-L.Z.); 14jzhang1@stu.edu.cn (J.L.); 14hzhuo1@stu.edu.cn (J.-H.Z.)

**Keywords:** chronic subdural hematoma, MMP, proangiogenic factor

## Abstract

Chronic subdural hematoma (CSDH) is a common neurological disease that involves the collection of blood products in the subdural space. The progression of CSDH is an angiogenic and inflammatory process, but the multifactorial mechanisms underlying CSDH are still not fully understood. We aimed to identify one or more factors that may play an important role in the development of CSDH. We enrolled 83 patients with CSDH, including 17 postoperative patients, and analyzed 20 markers in the hematoma fluid and peripheral blood of each patient. Overall differential gene expression was examined to identify the representative markers. The concentration of MMP-8 was significantly lower in the postoperative group than in the preoperative group. The concentration of MMP-9 was significantly higher in the postoperative group than in the preoperative group. These findings indicate that MMP-8 and MMP-9 may play important roles in the pathophysiology of CSDH. Understanding the pathways associated with CSDH may provide insights for improving disease outcomes.

## 1. Introduction

Chronic subdural hematoma (CSDH) is a common neurological disease that involves the collection of blood products in the subdural space. It has an indolent course of progression, usually spanning more than 3 weeks [1,2]. The reported incidence of CSDH ranges from 1.72 to 20.6 per 100,000 persons per year; however, for those aged 70 years or older, the incidence is estimated to be 58 per 100,000 [2,3]. Surgery is the first choice of treatment for CSDH with significant symptoms, which may involve burr-hole drainage (BHD), twist-drill craniotomy (TDC), or mini-craniotomy [3]. In addition, for patients with high surgical risk, non-surgical treatment has been reported to be safer and more effective [3]. Indeed, one study reported that atorvastatin combined with low-dose dexamethasone may play a significant role in the treatment of CSDH [4]. 

While CSDH is often associated with traumatic head injury, the development of CSDH is still poorly understood [2,3]. Most studies have investigated interrelated mechanisms implicated in CSDH, such as inflammation, membrane formation, angiogenesis, and fibrinolysis [1,5]. In a rat model of CSDH, the profile of inflammatory cytokine expression was elevated in the hematoma membrane and the surrounding tissue [6]. The affected cytokines included pro-inflammatory cytokines such as interleukin-6 (IL-6), interleukin-8 (IL–8), and tumor necrosis factor α (TNF-α); anti-inflammatory cytokines such as interleukin-10 (IL-10) and interleukin-13 (IL-13); and proangiogenic factors such as vascular endothelial growth factor (VEGF) and matrix metallopeptidase 9 (MMP-9) [6,7]. Tsukasa et al. suggested that IL-10 acts as a deactivator of inflammation in the pathogenesis of CSDH [8]. Atorvastatin can improve the level of endothelial progenitor cells and angiogenic factors in the peripheral blood, thereby promoting angiogenesis and the formation of functional blood vessels by activating the Akt pathway, Notchl pathway, and endothelial nitric oxide synthase. Indeed, previous research indicated that atorvastatin may improve prognoses in patients with CSDH [9]. Jun et al. reported that the VEGF concentration in the hematoma alone is not a reliable predictive biomarker for the natural history of CSDH or its recurrence. Another study indicated that the VEGF concentration decreases over time in patients with the trabecular type of CSDH [10]. Satoshi et al. further observed that cathepsin K levels in the peripheral venous blood are elevated in patients with CSDH, suggesting its potential role in CSDH development [11]. Brain natriuretic peptide (BNP) concentrations are also frequently elevated in CSDH and were reported to significantly increase after surgery, following which they return to the initial level [12]. Preoperative plasma BNP may also be an independent predictor of functional outcomes at follow-up [13]. In their study of neurotrauma associated with CSDH, Piers et al. [14] observed elevated serum levels of S-100b in patients with CSDH, although levels were significantly higher in the hematoma fluid. Nonetheless, the authors concluded that S-100b cannot be used as a biomarker for trauma or functional outcomes in patients with CSDH [14].

However, there is still little evidence regarding the factors influencing CSDH formation in humans [6,7,10]. We aimed to identify one or more factors that may play an important role in the formation of CSDH by examining a series of cytokines in CSDH and in the peripheral serum of affected patients.

## 2. Materials and Methods

### 2.1. Patients

We analyzed data for 83 patients with CSDH (76 men and 7 women) who had been admitted to the neurosurgery department of Shenzhen Second People’s Hospital from January 2019 to June 2021. Patients who were diagnosed with CSDH via computed tomography or magnetic resonance imaging and underwent surgical treatment were included. The exclusion criteria were as follows: history of other neurological diseases such as hydrocephalus or brain cancer, obvious bacterial infection, long-term use of thrombolytics or anticoagulant drugs, and treatment with anti-inflammatory agents such as penicillin prior to admission. All patients had severe symptoms of headache or drowsiness due to CSDH and surgery. All patients were treated with a single burr-hole drain without irrigation under general anesthesia. After surgery, all the patients were treated with atorvastatin and dexamethasone. Among them, we followed up 17 patients who had undergone BHD surgery. Hence, the preoperative CSDH group included all 83 patients, while the postoperative CSDH group included the 17 patients who had undergone follow up. This clinical study was approved by the Ethics Committee of the Shenzhen Second People’s Hospital. (20200422003-XZ2021-XZ2021). Informed consent was obtained from all patients. 

### 2.2. Cytokine Measurements

Peripheral blood samples were collected from each patient 30 min before surgery prior to and 1 month after the surgery. Fluid samples were also collected from the hematoma during BHD surgeries. All samples were collected in tubes containing a coagulator and were immediately centrifuged at 2000 rpm for 15 min. After centrifugation, the supernatants were stored in sealed polypropylene tubes at −80 °C until further analysis.

The samples of hematoma fluid and the preoperative and postoperative peripheral blood samples were analyzed using a 20-plex human panel A system (R&D Systems, Minneapolis, MN, USA), a Luminex system (Luminex, Austin, TX, USA), and Bioplex software (BioRad, Hercules, CA, USA). This panel evaluates MMP-1, MMP-2, MMP-3, MMP-8, MMP-9, platelet-derived growth factor-BB (PDGF-BB), angiopoietin-2, monocyte chemoattractant protein-1 (MCP-1), D-dimer, epidermal growth factor (EGF), hepatocyte growth factor (HGF), insulin-like growth factor binding protein-3 (IGFBP-3), IL-1α, IL-6, IL-8, IL-10, prolactin, TNF-α, VEGF, and vascular endothelial growth factor receptor-2 (VEGFR2). 

### 2.3. Differentially Expressed Gene (DEG) Screening

We assessed overall differential gene expression using the ggplot2 package (http://ggplot2.org, accessed on 21 August 2021) in R statistical software (version 4.1.0, The R Foundation for statistical computing, Auckland, New Zealand). Further, we used the LIMMA package to select the DEGs. The empirical Bayes moderated *t*-test method was used to calculate the *p*-values for each cytokine. We also calculated the adjusted *p*-values based on the false discovery rate (FDR). Only genes with a log_2_ fold change (FC) > 2 and FDR < 0.01 were considered DEGs.

### 2.4. Statistical Analysis

Statistical analyses were performed using the Mann–Whitney *U*-test. Data are presented as the mean ± standard deviation. All analyses were performed using SPSS software (version 25, IBM Corporation, Armonk, NY, USA). A *p*-value < 0.05 was considered statistically significant.

## 3. Results

### 3.1. Baseline Patient Characteristics

The mean age in the preoperative CSDH group was 66.73 ± 15.14 years; most patients in this group were male (91.6%). The general characteristics of the groups are presented in Table 1.

### 3.2. Identification of DEGs 

All samples of hematoma fluid and peripheral blood, obtained preoperatively and postoperatively, were analyzed. An overview of differential gene expression is presented in Table 2 and Table 3.

Our analysis indicated that VEGF, PDGF-BB, TNF-alpha, MMP-9, IL-6, CCL2, IL-1 alpha, MMP-1, MMP-8, and IL-8 were DEGs between the hematoma fluid and the peripheral blood before surgery. An overview of the differential expression of all genes is presented in Table 3. MMP-2, PDGF-BB, HGF, MMP-9, EGF, MMP-8, D-dimer, and prolactin were identified as DEGs between the preoperative and postoperative peripheral blood samples. 

### 3.3. Analysis of the Data

The concentration of MMP-8 was significantly lower (*p* < 0.01) in the postoperative group than in the preoperative group, while the concentration of MMP-9 was significantly higher (*p* < 0.01) (Table 4).

## 4. Discussion

CSDH is a common disease in older adults that can be treated surgically. However, many patients experience recurrence of CSDH after surgery, the rate of which ranges from 2% to 37% [15]; hence, understanding the pathogenesis of CSDH remains essential. The progression of CSDH is known to be an angiogenic and inflammatory process, but the multifactorial mechanisms underlying CSDH are still not fully understood [16]. In addition, fibrinolysis has been reported to play a role in CSDH formation [17]. Many inflammatory cytokines and angiogenic factors have been assessed in patients with CSDH, including IL-6, IL-8, IL-10, transforming growth factor β (TGF-β), VEGF, MMP-9, and others [16,17,18,19]. However, the role of these factors in hematoma formation remains unclear. In this study, we investigated the role of 20 cytokines in the inflammation and angiogenesis related to the formation of CSDH. Our findings indicated that the concentrations of most of the cytokines evaluated changed in the hematoma fluid, indicating that they may be involved in CSDH formation. The concentration of MMP-8 was significantly lower in the postoperative group than in the preoperative group, while the concentration of MMP-9 was significantly higher. Thus, these two cytokines may contribute to the pathogenesis of CSDH in different ways.

MMPs are a family of zinc-dependent proteolytic enzymes, and their expression is elevated in most diseases that involve inflammation [6]. They play a critical role in vascular formation and remodeling. The increased MMP activity in some pathophysiological conditions may lead to pathological alterations in blood vessels [20].

MMP-8 was reported to play an important role in several pathological conditions, especially in inflammatory conditions such as heart disease, osteoarthritis, and multiple sclerosis [21]. While some studies have analyzed the role of MMP-8 in CSDH, they did not especially focus on it [20,22]. In our study, we observed that the concentration of MMP-8 was higher in hematoma fluid than in the preoperative peripheral blood, decreasing following treatment. We thus inferred that MMP-8 is involved in CSDH formation. Lower postoperative MMP-8 levels may represent a marker for evaluating the prognosis of CSDH. Future studies should analyze the associations between MMP-8 concentration in preoperative/postoperative serum and hematoma fluid to elucidate its role in CSDH formation.

Many studies have investigated the role of MMP-9 in the pathophysiological pathways of CSDH [5,19,20,23], suggesting that it plays a significant role in blood vessel formation and vascular remodeling [20]. MMPs, along with VEGF, contribute to the instability of newly formed blood vessels, which can lead to a higher risk of hemorrhage [5]. In rats, MMP-9 has been reported to correlate with VEGF concentration in the external neomembrane and promote leaky blood vessel formation and subsequent rebleeding [7]. MMP expression accelerates inflammatory processes by modulating other mediators, such as cytokines and chemokines [5]. Furthermore, studies have indicated that MMP-9 reduces the absorption of CSDH due to increased vascular permeability, enhanced inflammation, and reduced vascular maturation [19]. Some investigators have provided explicit evidence that the concentrations of MMP-2 and MMP-9 are significantly elevated in hematoma fluid, indicating that the MMP/VEGF system may be involved in the angiogenesis associated with CSDH [24]. In contrast, we observed a reduced concentration of MMP-9 in the hematoma fluid. This may indicate that the hematoma cavity contains few blood vessels, and that the hematoma develops from newly formed unstable blood vessels. In contrast, elevated levels of MMP-9 in the postoperative serum may promote normal angiogenesis, which can aid in the recovery for patients with CSDH. As the pathogenesis of CSDH has emerged, researchers have also focused on developing appropriate treatments. Fu et al. noted that atorvastatin may enhance angiogenesis and reduce CSDH-related inflammation [25]. Furthermore, Wang et al. observed that atorvastatin combined with low-dose dexamethasone was more effective than atorvastatin alone in reducing hematoma volume and improving neurological function in patients with CSDH [4]. Fan et al. also reported that combination therapy with atorvastatin and low-dose dexamethasone counteracted hematoma-induced Kruppel-like factor-2 suppression in human cerebral endothelial cells, thus reducing endothelial inflammation and permeability [26]. There is further evidence indicating that atorvastatin promotes angiogenesis and reduces inflammation, both of which are associated with the formation of CSDH [27,28]. Araújo et al. demonstrated that the inhibitory function of atorvastatin is associated with the modulatory effects of HMG-CoA reductase on VEGF, TNF-α, and TGF-beta 1 production [27]. Since the concentration of MMP-9 in the peripheral blood is higher postoperatively than preoperatively, it may play an important role in angiogenesis post-treatment. Combined treatment with atorvastatin and dexamethasone may help to inhibit inflammation and promote the formation of new blood vessels. In addition, the concentration of MMP-9 was elevated, suggesting why atorvastatin or dexamethasone can increase the expression of MMP-9.

CSDH can be treated using multiple approaches, but recurrence is the major complication requiring further treatment such as reoperation [29]. The relevant literature indicates that the rate of reoperation ranges from 10% to 20% [29]. Early prevention can aid in controlling the recurrence rate among patients with CSDH. Kim et al. analyzed the degree and density of CSDH on computed tomography and found that recurrence rates were higher for single-layer or isodense hematomas [30]. The degree or density of different layers of CSDH may further be associated with the concentrations of MMP-8 and MMP-9. Inflammatory markers in hematoma fluid, including IL-6 and IL-8, were shown to predict CSDH recurrence [31]. Our findings indicate that MMP-8 and MMP-9 may serve as accessible markers for monitoring the progress of CSDH and its prognosis after surgery. Further, MMP-8 and MMP-9 may exert effects on one another. Although evidence suggests that atorvastatin or dexamethasone can inhibit the expression of MMP-8, further studies are required to confirm this hypothesis. Future studies may also wish to examine serum concentrations of these cytokines in relation to prognosis and prevention of recurrence in patients with CSDH.

This study had some limitations. Some patients could not commute to our hospital to complete the experiment and instead underwent the CT reexamination in a local hospital, while still others could not be contacted. Further, the number of patients included in the study was relatively small, meaning that the study population may not accurately represent the wide spectrum of patients with CSDH. Further studies involving larger sample sizes, especially postoperatively, are required to verify our findings. Furthermore, to confirm the role of MMPs in neuroinflammatory processes, future studies should investigate the specific cytokine profiles in the outer membranes of CSDH. Multiple factors, including MMP-8, MMP-9, and other interacting clinical and biochemical mediators, are involved in the formation of CSDH. As such, additional research is required to identify these factors and the role of their interaction in CSDH.

## 5. Conclusions

The current study investigated cytokines in hematoma fluid and peripheral venous blood samples from patients with CSDH. Our analysis revealed significant differences in the preoperative and postoperative levels of MMP-8 and MMP-9 among the included patients. Thus, these two cytokines may play an important role in the pathophysiology of CSDH and may represent useful markers for treatment evaluation and follow-up. Thus far, our results provide interesting insights into the biochemical processes involved in the development of CSDH. Further animal and clinical studies are required to elucidate the potential of MMP-8 and MMP-9 as indicators for monitoring CSDH, evaluating therapeutic effects, and preventing recurrence.

## Figures and Tables

**Table 1 jcm-11-00902-t001:** Age and sex in each group.

	Age	Gender	
		Male	Female
Preoperative CSDH group	66.73 ± 15.14	76	7
Postoperative CSDH group	62.63 ± 15.17	17	0

**Table 2 jcm-11-00902-t002:** d = Differential expression of all cytokines between the hematoma fluid and preoperative peripheral blood samples.

Cytokine	logFC	AveExpr	t	*p* Value	adj.*p*.Val	B
VEGF	9390.762	4869.77	11.44981	2.92 × 10^−22^	3.80 × 10^−21^	−4.58994
PDGF-BB	−3476.56	2598.961	−11.1052	2.45 × 10^−21^	1.59 × 10^−20^	−4.59011
TNF-alpha	396.2944	239.1504	9.328201	1.24 × 10^−16^	5.39 × 10^−16^	−4.59106
MMP-9	−172,297	131,672.2	−9.27536	1.71 × 10^−16^	5.55 × 10^−16^	−4.59109
IL_6	12,141.2	6071.305	9.09912	4.89 × 10^−16^	1.27 × 10^−15^	−4.59119
CCL2	16,007.98	8430.163	7.747179	1.26 × 10^−12^	2.72 × 10^−12^	−4.59198
IL-1 alpha	246.7596	147.8462	7.499382	5.03 × 10^−12^	9.34 × 10^−12^	−4.59213
MMP-1	36,338.67	28,088.57	6.338925	2.52 × 10^−9^	4.09 × 10^−9^	−4.59282
MMP-8	39,584.56	29,068.01	5.765987	4.41 × 10^−8^	6.37 × 10^−8^	−4.59316
IL-8	8246.851	4156.585	5.070263	1.14 × 10^−6^	1.49 × 10^−6^	−4.59355

logFC: differential expression multiple; AveExpr: mean expression level; t: T value (a paired-samples *t*-test); adj.*p*.Val: adjusted *p* value; B: the logarithm of the standard deviation after Bayes analysis.

**Table 3 jcm-11-00902-t003:** Differential expression of all cytokines between the preoperative and postoperative peripheral blood samples.

Cytokine	logFC	AveExpr	t	*p* Value	adj.*p*.Val	B
MMP-2	966,877.4	468,478.9	22.09931	2.03 × 10^−38^	3.66 × 10^−37^	−4.47072
PDGF-BB	11,225.27	6257.109	15.59472	1.75 × 10^−27^	1.57 × 10^−26^	−4.48801
HGF	−288.401	397.9733	−3.04988	0.002996	0.014214	−4.58293
MMP-9	141,607.5	241,620.4	3.032322	0.003159	0.014214	−4.58307
EGF	−166.278	258.8795	−2.71809	0.007857	0.028285	−4.58556
MMP-8	−6401.36	8303.789	−2.51726	0.013571	0.040712	−4.58705
D-dimer	−2,243,488	3,945,004	−2.37019	0.019883	0.048286	−4.58808
Prolactin	−25,199.5	28,945.43	−2.34004	0.02146	0.048286	−4.58829

logFC: differential expression multiple; AveExpr: mean expression level; t: T value (a paired-samples *t*-test); adj.*p*.Val: adjusted *p* value; B: the logarithm of the standard deviation after Bayes analysis.

**Table 4 jcm-11-00902-t004:** Concentration of MMP-8 and MMP-9 in the serum and hematoma fluid in patients with CSDH (*p* < 0.01).

Factor	MMP-8 (ng/mL)	MMP-9 (ng/mL)
Preoperative serum	9.18 ± 9.78	217.19 ± 155.02
Hematoma fluid	78.24 ± 216.30	45.33 ± 52.47
Postoperative serum	2.97 ± 3.80	354.13 ± 231.55

## Data Availability

Data can be accessed by contacting the corresponding author on reasonable request.

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
