# Peer review of "Serum Levels of MMP-8 and MMP-9 as Markers in Chronic Subdural Hematoma"

_jcm, 2022, doi:10.3390/jcm11040902_

Round 1
Reviewer 1 Report
In this study, the authors aim to assess levels of MMP-8 and MMP-9 in serum and hematoma fluid in patients with cronic subdural hematoma (CSDH). While such an approach bears the potential to better understand and predict CSDH formation, the reviewer has several major concerns that render this manuscript not suitable for publication in MDPI JCM:
CSDH is a frequent condition in the elderly. By contrast, the number of patients in this study, particularly in the post-operative cohort, is very low, and the male to female ratio in the whole cohort (approx. 11:1) is far too high as compared to other clinical studies on CSDH. These highly biased demographic characteristics render the results of the present study not convincing.
The authors found a handful of differentially expressed genes (DEGs) in blood serum and hematoma fluid samples. While this result is statistically significant, it proves no more than the fact that blood serum and CSDH fluid are distinct liquids, which is self-evident.
Minor concerns:
It is not clear from the manuscript if the same gene panel was used for the screening of blood samples and hematoma fluid samples.
The authors claim to have used the R ggplot2 package to assess part of the data, e.g. DEGs. This R package usually returns figures that are a feast for the eyes. So, how about adding some figures to the manuscript?
Reviewer 2 Report
Serum levels of MMP-8 and MMP-9 as markers in chronic subdural hematoma
This article presents findings of two specific enzymes in patients with chronic subdural hematoma (CSDH), both in serum and in the hematoma fluid. In some patients, the serum levels were measured both before and after surgery. The enzymes, MMP-8 and MMP-9, are both so-called collagenases, involved in the breakdown of collagen. It is an interesting experiment, but the article seems unfinished, and in some ways quite confusing.
Abstract
Line 13: "representative markers of ... prevention". I do not think you can justify that based on the study design.
Line 15: why did you do a gene analysis?
Introduction
Line 38: interleukin-6 is mentioned twice.
Materials and Methods
Line 62: Serum samples were taken preoperatively. I think it would be possible, and more accurate, to draw both serum samples and hematoma fluid samples at the same time, i.e. during surgery.
Line 77: I am really confused. Why did you analyze genes? And what genes were analyzed? You are in this section partly discussing genes and partly discussing cytokines. You have to tidy this up.
Line 84: Statistics. Again, are you referring to cytokine analysis or genes? Regarding cytokines before and after surgery: did you consider a paired-samples t-test?
Results
Line 90: Very little information about the patients is presented. It would be interesting to see if for example symptom severity or duration is associated with the cytokine levels.
Line 101: Table 2. This table is impossible to interpret. How do I see levels in the hematoma versus serum? What does the letters mean? For instance, what is column B?
Line 110: Table 3. Same as Table 2. At least you need some footnotes to explain the letters.
Line 112: You just mentioned several cytokines differing between preoperative and postoperative serum, yet you only discuss two (MMP-8 and MMP-9). This seems strange. Why?
Discussion
Again I miss a discussion regarding specific patient types. Will the symptoms or duration of symptoms play a role? Many old people with CSDH use antiplatelet/anticoagulant medicines – how will this influence the results? Where they all trauma patients?
About MMP-8 and MMP-9: Does it exist normal serum values? Could the surgical trauma itself cause changes in serum levels? It would also be interesting if you could measure MMP levels in CSDH patients not available for surgery – how do these levels change during the disease? To put it in other words: are your results logical?
Line 232: it says «NAME OF INSTITUTE»…
Reviewer 3 Report
Dear Authors,
Thank you very much for submitting the artcile. However, there are several minor and major concerns that should be addressed.
- Introduction should be more focused on biomarkers of cSDH.
- Important literature missing: 10.1007/s10143-006-0019-7, 1007/s10143-019-01218-w, 10.1016/j.clineuro.2020.106458, 10.1016/j.jocn.2017.08.021
- Similar research should also be discussed: 10.1016/j.jns.2020.117240, 10.1089/neu.2020.7110, 10.1016/j.jocn.2019.05.058
- What is mend with the abbreviation of BHD?
- Why did you collect blood samples of 83 patients, while you only collected fluid samples of 17 patients? Statistical analysis of only 17 patients is poor.
- How did you follow up those patients?
- Did you correlate the blood samples and the intraoperative samples with volume of cSDH? If not, please do so.
- Discussion should focus on the results and the differences with the current literature. Additionally, authors should discuss other serum markers/biomarkers presented in literature (please also see point 2 and 3).
- Limitations have to be pointed out in detail.
Reviewer 4 Report
The manuscript has the classic structure of a research article.
Tables are legible and properly signed.
The inclusion and exclusion criteria should be expanded.
I don't know which neurological diseases and anti-inflammatory drugs
were included in the exclusion criteria. There is no information on
how long before and after the operation, peripheral blood is collected
for testing.
In addition, it is also necessary to specify what type of surgery was
performed on the patients and whether each of them was treated with the
same method.
The discussion is comprehensive and logical. The conclusions are clear.
Round 2
Reviewer 1 Report
Major concerns have been addressed adequately.
Reviewer 2 Report
Thank you for the revised manuscript. I have no major objections of it being published.
Reviewer 3 Report
Dear Authors,
Thank you very much for adopting most of the major concerns of each reviewer. The quality of the article improved likewise. However, key limitation is the amount of patients included (17 patients). Nevertheless it remains an interesting topic
This manuscript is a resubmission of an earlier submission. The following is a list of the peer review reports and author responses from that submission.